# Is Public Health Response to the Phenomenon of Alcohol Use during Pregnancy Adequate to the Polish Women’s Needs?

**DOI:** 10.3390/ijerph19084552

**Published:** 2022-04-09

**Authors:** Katarzyna Okulicz-Kozaryn

**Affiliations:** 1Institute of Mother and Child, 01-211 Warsaw, Poland; katarzyna.okulicz@imid.med.pl; 2National Centre for Prevention of Addictions, 02-776 Warsaw, Poland

**Keywords:** fetal alcohol spectrum disorders (FASD), qualitative data, women’s perspective, public health response

## Abstract

Due to the risks it poses to a child’s health, drinking alcohol during pregnancy is a serious problem that the public health sector is struggling to deal with. The reasons why women who do not have alcohol problems do not give up drinking alcohol completely during pregnancy are still poorly understood. And the knowledge available about them does not translate into communication strategies in Poland. The analysis of standards and examples of good practice allows to formulate proposals for improving the quality and effectiveness of social campaigns addressed to the general population and women of childbearing age in order to reduce the risk associated with the prenatal exposure to alcohol.

## 1. Introduction

Drinking alcohol is one of the health behaviors about which every person makes countless decisions throughout their life. When deciding whether to drink or not to drink, what, in what amounts, when, in what company, etc., people rely on their previous experiences, behaviors observed in other people, normative beliefs, knowledge about the consequences [1,2]. Each time, of great importance are factors related to the context in which the decision is made, including the temporary emotional state (e.g., [3,4,5,6]). In many cases, drinking alcohol creates a conflict between what people can and want to do and what they should do for health, social or legal reasons. Dilemmas usually involve the effects of alcohol on the drinker’s own health.

However, for women who are pregnant or planning to become pregnant, the key issue is the effects of alcohol not on her own, but on the health of the baby to be born [7]. Consumption of alcohol during pregnancy can result in a number of adverse consequences for the fetus, including congenital malformations and behavioral, cognitive and adaptive deficits. Animal studies have shown that each stage of embryonic development is susceptible to the teratogenic effects of alcohol [8]. Prenatal exposure to alcohol is a major cause of brain damage and developmental delay known as Fetal Alcohol Spectrum Disorder (FASD). FASD patients may have up to 428 different medical conditions that cause serious complications throughout their lives [9].

How women make decisions about drinking alcohol during pregnancy is not fully understood. It is known that most women (70–87%) give up alcohol as soon as they find out that they are pregnant [10,11]. It is also known that for women addicted to alcohol, who, for instance, in USA constitute about 5% of the pregnant population [12], making such a decision is too difficult. But what about women that are not addicted, know they are pregnant, and still drink alcohol as before, or only limit the frequency and/or amount of alcohol consumption?

The first aim of this article is to analyze the current state of knowledge about the reasons for drinking alcohol, especially moderate drinking, during pregnancy. The considerations are based mainly on the results of foreign and Polish qualitative research. The second goal is to look at the activities in the field of public education carried out in Poland so far, in terms of their adequacy to the needs in the field of prevention of prenatal alcohol exposure (PAE) and fetal alcohol spectrum disorders (FASD). Combining the knowledge about how women think about drinking alcohol during pregnancy with the current practice and knowledge about effective interventions through the media will allow to formulate recommendations for future activities in this area.

The Polish context offers an interesting environment to study the phenomenon of public health response to alcohol use during pregnancy. Firstly, the estimated prevalence of alcohol consumption by pregnant women and FASD in Poland are similar to the situation in other countries of the European region [7,13]. Secondly, Poland may be regarded as a representative of Eastern European countries, where alcohol control systems integrated within the state-controlled economy were rapidly replaced (in early 1990s) by a laissez-faire approach and high alcohol consumption. Still, the level of alcohol control policies in this region is classified as medium or low and dominating public attitudes towards alcohol policies are supportive for laissez faire, including perception of alcohol as ordinary commodity and support for individual responsibility to protect yourself against alcohol harm [14]. And third, in the scientific literature there are very few reports of drinking and pregnancy in Eastern European countries, so the topic is poorly understood.

## 2. Understanding Women’s Reasoning about Alcohol Use during Pregnancy

The literature (e.g., [15,16,17,18,19]) indicates many different factors that increase the risk of drinking alcohol during pregnancy, including: lack of knowledge, medical conditions (e.g., addiction, mental disorders) or contextual factors (e.g., trauma, domestic violence, partner’s influence). However, from the perspective of public health responses to the challenges arising from the fact that the prevalence of alcohol use during pregnancy is rather not diminishing [20], better understanding of women’s reasoning about alcohol use in pregnancy is needed. Gaining insight into women’s beliefs and understanding of risk associated with alcohol use during pregnancy is necessary to inform targeted prevention strategies. Unfortunately, there is the paucity of studies exploring women’s perspective on alcohol use during pregnancy. Systematic review [21] identified only 27 qualitative studies published in the past 30 years (1990–2020) on factors influencing alcohol use in pregnancy. And only 7 of these studies were conducted in Europe (France, The Netherlands, Switzerland, Sweden and UK).

From these European studies we know that women’s perception of the risk associated with drinking alcohol during pregnancy vary greatly due to inconsistent information that reaches them from various sources, including government guidelines, health organizations, media, family and friends. Moreover, women rarely receive individualized advice from health professionals [22,23]. The information on the impact of alcohol use on a fetus is often misunderstood by women [24,25]. Although smoking during pregnancy is generally considered to be a risk-taking behavior, moderate drinking of alcohol, in small amounts, is perceived as acceptable [25,26]. Alcohol use by a partner, his encouragement of light drinking or support in handling the struggles associated with reducing alcohol intake, is a significant factor determining alcohol use in pregnancy [23,26,27,28]. Women who continue to drink during pregnancy usually do so in social contexts of festive occasions [28]. As shown in the study comparing practices and attitudes towards alcohol use during pregnancy in England and Sweden “wider social norms and attitudes, interlinked within the policy context, may influence whether pregnant women drink alcohol” [28]. It can also be assumed that the differences found in the UK and Swiss studies in the importance of reducing the risk of drinking alcohol during pregnancy ascribed to health professionals reflect cultural (national) differences. In Switzerland health professionals were rarely mentioned as important resources of support [27] while in UK support from health services played an important role in shaping women’s behavior [26].

## 3. Exploration of Polish Women’s Reasoning about Alcohol and Pregnancy

Interesting qualitative study among Polish women, commissioned by the State Agency for Prevention of Alcohol Related Problems (PARPA), was conducted by the 4P Research Company [29]. It aimed to explore women’s knowledge, attitudes and believes towards alcohol consumption during pregnancy. Participants of focus group interviews (FGI) were recruited in Warsaw (capital city) and Płock (smaller town) from the general population of women (with various education, occupation and parenting status) of childbearing age (24–39 years old). Half of respondents was pregnant and half not pregnant, but not ruling out the possibility of becoming mothers in the foreseeable future. All non-pregnant women were not abstainers currently and pregnant women were not abstainers before pregnancy. Half of the respondents rejected alcohol use during pregnancy while the others did not (thought that some alcohol during pregnancy is acceptable). In order to facilitate open conversation the composition of each focus group was based on participants pregnancy status (yes/no) and attitudes toward alcohol use during pregnancy (not acceptable/acceptable) (Table 1).

FGI were conducted by the experienced moderator with groups of 6 respondents and each of them lasted about 2 h. To stimulate discussion and facilitate elucidation of concepts that are important for respondents visual material were used. Associations with pregnancy and methods of taking care of the pregnancy were activated by photos (of food, activities, landscapes, social situations, objects, etc.), from which each respondent chose the ones that were most associated with the topic of conversation. The second set of photos were “portraits” of women and the task of the respondents was to identify those who during pregnancy may regularly drink alcohol, who may occasionally use alcohol, and those who strongly reject any alcohol during pregnancy.

The key findings are summarized below. The issues already discussed in European qualitative studies (mentioned above) are omitted (e.g., contradicting information from different sources, lack of knowledge on FASD and consequences of fetal alcohol exposure, greater harm attributed to smoking than drinking alcohol, insufficient involvement of medical staff in FASD prevention (doctors rarely raise the topic of drinking alcohol during pregnancy themselves, and some allow occasional drinking for medical (e.g., regulation of blood pressure, improving the work of the kidneys) or socio-emotional (having fun in a company) reasons.:

Key findings: Being a good mother is equally important for all women, regardless of their attitudes and behaviors regarding alcohol consumption during pregnancy. Women accepting (small amounts of lighter) alcohol during pregnancy seem just as committed (future and current) mothers as those who completely reject drinking alcohol during pregnancy. They associate pregnancy with healthy lifestyle, family, time of change and expectation.Locus of control over the pregnancy (me versus my doctor).
“First of all, do no harm”—characterize the dominating approach toward care of the unborn child and pregnancy of women definitely rejecting alcohol during pregnancy. Not surprisingly, this harm avoidance perspective includes alcohol abstinence and makes the mother the key health agent.“Medical” factors are crucial—women accepting small amounts of alcohol during pregnancy underlined the importance of good medical care (regular visits to the doctor and adherence to recommendations, ultrasound examination, etc.). This suggests a shift in the responsibility for the outcome of pregnancy from the mother to the medical staff.No evidence of harmful effects of moderate drinking has different meaning for women accepting or rejecting small amounts of alcohol during pregnancy. In general, an opinion prevails that there is no clear evidence that small amounts of light alcohol are harmful to the foetus, so:
For women who accept drinking alcohol during pregnancy, this means that there is no evidence of harmful effects and you can drink (a little).For women who do not accept drinking alcohol during pregnancy, this means that there is no evidence of harmlessness, so you can’t drink.“FAS (fetal alcohol syndrome) occurs in pathological families”. Among women who accept drinking during pregnancy, there is a strong emotional distancing from the threat of FAS by identifying it (only) with the children of mothers who are alcoholics, drink compulsively or get drunk regularly. This seems to reflect media “bombshells” (for example about “drunk newborns”).“I wouldn’t tell the doctor” for fear of his/her reaction. Women think that it is easier to talk with a friend or write in internet about alcohol and pregnancy than to talk openly with a doctor. They expect criticism, e.g., “Oh, what have you done!?” or disregard, as one respondent said: “It’s if I asked about the FAS, she [the doctor] would say to me: “Madam, madam … —she would look at me—No, this does not apply to you.” Psychologist and fortune teller in one”. It should be noted that there were also respondents who, although did not deny the general opinion about doctors, emphasized that they are extremely lucky because they can talk to their doctor about everything.Mental rationalizations among women who accept alcohol consumption or drink alcohol during pregnancy to convince or reassure themselves in respect of threats to the child related to prenatal alcohol exposure:
“Pregnancy is not a disease”—this very well known in Poland slogan is interpreted as encouragement to maintain in pregnancy the same lifestyle (including alcohol use in moderate amounts) as before.There are other, not avoidable teratogens-e.g., air pollution, some medicines with unknown impact on fetus.Alcohol abstinence during pregnancy is a „new-fashioned” exaggeration and dictatorship of bans and ordersPregnant women are constantly threatened and criticized—„No matter what I do, it will be considered as wrong”Medical recommendations (in general) are not stable in time and probably those regarding alcohol use in pregnancy would be changed soon.

This unique in Poland study revealed many different opinions, myths, misunderstandings and approaches toward alcohol use during pregnancy. It made clear that in spite of several efforts to disseminate information on FAS/FASD in the public space, women’s knowledge is fragmentary and imprecise and therefore, they are not sure how to behave.

## 4. Opinions and Expectations about Information Campaigns on Drinking Alcohol and Pregnancy

Voices of women participating in the focus group interviews [29] indicated that communication strategies addressing women who do not reject drinking or drink moderately during pregnancy are needed. Especially for women before and during their first pregnancy, when the level of anxiety and uncertainty is higher than during the next one. These women care about the health of their children, and when it comes to alcohol, they are not completely sure if they are doing the right thing. The following guidelines for communication planning follow from the study:Avoid criticizing or stigmatizing alcohol consumption during pregnancy, because pregnant women/mothers feel beleaguered by social criticism, so they often respond to it with some kind of defiance resulting from helplessness.Avoid excessive “scaring”—because pregnant women/mothers feel “constantly threatened” and scaring can give effects opposite to the intentions and cause rejection of threatening information.What may have the potential is showing real life stories (testimonials), which document that FASD can happen in a “normal” family, where the mother is not an alcoholic, but drank moderately during pregnancy or before knowing she is pregnant. Presentation of a case-a child with FASD would have high emotional load and therefore may have a tremendous influence on one’s attitude towards drinking during pregnancy. Due to emotionality and rooting in everyday life, this communication direction should also be effective in case of people who rely rather on tradition and life experience than “science”.Since there is no evidence of the harmful effects of exposure to small amounts of alcohol on the fetus, the campaigns should reveal the irrationality of the applied mental strategies (e.g., “If you saw someone crossing the street outside the crosswalk and not being run over, does it mean that such crossing is not associated with risk?”).The message should be simple, clear and not leaving any room for different interpretations. As a good example may serve campaigns on drink driving, in which nobody discuss that a small amount of alcohol in the blood is acceptable. Similarly, in reference to pregnancy any doubts concerning different harmfulness of prenatal alcohol exposure due to individual differences between woman or the stage of pregnancy should be omitted.The best remembered are “strong” messages, exemplified by social campaigns against cigarette smoking during pregnancy, as cited by one of the respondents: “Do not turn your belly into a gas chamber”.Communication channels that may have potential are for example blogs (mothers-bloggers are perceived as close and reliable). Besides, also midwives could be educators for women (perceived as being nearer to “real life” and thus more reliable than doctors).

## 5. Public Awareness Campaigns on FAS/FASD in Poland

It is interesting to look at the public awareness campaigns implemented so far in Poland from the perspective of presented above, opinions of the representatives of the target population—women in childbearing age, not abstaining from alcohol (at least before getting pregnant). Some past campaigns aimed at FASD prevention implemented in Poland are described in [30]. Unfortunately, none of the older or newer campaigns have been assessed in the evaluation studies, so it is not known whether they had any influence on the behavior of women during pregnancy. We can only analyze their concept, content and forms.

Probably the most influential, especially among local stakeholders, has been “Ciąża bez alkoholu” [“Pregnancy without alcohol”] initiated by PARPA (State Agency for Prevention of Alcohol-Related Problems) in 2007. It managed to engage various institutions, local authorities and NGOs and significantly contributed to the spread of awareness of the harmful effect of alcohol consumption during pregnancy on the health of an unborn child. The campaign’s slogan “I don’t drink to his health” referred to the most common Polish toast: “To health!”. By definition, the campaign was positive, did not stigmatize pregnant women and did not threaten with damages resulting from PAE. Instead, it showed a somewhat schematic, idealized image of a happy family that had little to do with real-life stories. Currently, the educational website with the same title (Pregnancy without alcohol) is available for general population and professionals (www.ciazabezalkoholu.pl, accessed on 16 February 2022).

Another campaign which has been running still from 2014 is “Stop FAS” implemented at the regional level (Pomeranian Voivodship). The recent activities include videos e.g., showing a women refusing a drink because of pregnancy (Spot: “Dlaczego ona nie pije?” (“Why is she not drinking” https://kampaniastopfas.wixsite.com/stopfas/spoty, accessed on 16 February 2022). The information on boards says: “when you are drinking your child is drinking with you” and “even small amounts of alcohol can damage your baby’s brain, heart and other organs” summarized with the statement “I don’t drink when pregnant”. This message seems to be unambiguous, understandable and contains a clear indication of how to behave. The problem is that the scope of this campaign is minimal and probably only reaches people actively seeking information on media activities on FASD prevention (like the author of this article).

A person interested in the topic, can find in the internet other examples of preventive activities, e.g., educational campaign entitled “Mom, don’t hurt-FAS Prevention”. The campaign co-financed by the Marshal’s Office of the Małopolska Region was recognized as the best social project in 2019, and its coordinator–Ad vocem foundation received the “Salt Crystal” award in the category: Best Social Project (https://www.advocem.org.pl/2021/12/13/jak-przerwac-dramat-dzieci-krzywdzonych-copy-3-copy-copy-copy-copy-copy/ accessed on 16 February 2022). The tone of this campaign is completely different from the ones discussed above. It is part of a wider spectrum of activities of the Ad vocem foundation under the slogan “How to end the drama of abused children”. It stigmatize women drinking alcohol during pregnancy as purposefully harming their unborn children. It also evokes strong, negative emotions what probably may increase its recognition.

Currently, the most visible on the Internet is the FASOFF campaign sponsored by a brewing company, realized in the convention of a horror movie (https://www.fasoff.pl/, accessed on 16 February 2022). Moreover, this scaring form is explained by the Authors: “Anything on this site annoys you? We know. This is what the world looks like from the perspective of a child affected by FAS”. Actually, FASOFF is the only campaign mentioned in the interviews conducted by Golińska [27] although not explicitly (with no name/title). When asked if it was a convincing advertisement, the respondent replied with a clear hesitation: “I think so … If someone has no idea that there is such a disease … Although this FAS is more about addicts.” This suggest that the message is unlikely to reach the general population of women who do not have alcohol problems.

Based on the above examples, it may be concluded that the visibility of FASD prevention campaigns is very limited. Most women of childbearing age have never seen such activities at all. Although there are some examples of campaigns clearly avoiding scaring, criticizing or stigmatizing alcohol consumption during pregnancy, but there are also examples of activities based on the opposite approach. In all campaigns, the messages are rather simple and clear (“do not drink alcohol during pregnancy”). The emotional load of campaigns using threatening strategies is much stronger than in “positive” campaigns, but still, most of the addressees do not identify themselves with it, thinking that it is rather directed to people addicted to alcohol. There are no campaigns using testimonials or referring to the mental strategies used to justify moderate drinking during pregnancy.

## 6. (Dis) Accordance of Polish Public Awareness Campaigns on FAS/FASD with Quality Standards

International Standards on Drug Use Prevention [31] provide characteristics of media campaigns deemed to be associated with efficacy and/or effectiveness (pp. 33–34):“They precisely identify the target group of the campaign.They are based on a solid theoretical basis.The messages employed are designed on the basis of strong formative research.They strongly connect with other existing drug prevention programmes in the home, school and community.They achieve adequate exposure of the target group for a long period of time.They are evaluated systematically.They target parents, as this also appears to have an independent effect on the children.They are aimed at changing cultural norms about substance use, educating about the consequences of substance use and/or suggesting strategies to resist substance use.”

Unfortunately, none of the Polish campaigns aimed at reducing the risk of drinking alcohol during pregnancy and FASD meets the above criteria. In contrary, most of the campaigns are badly designed (without testing of their concepts) and implemented with limited resource (if not sponsored by the alcohol industry). Any of them has been evaluated and most of them passed unnoticed due to insufficient coverage, limited forms and implementation time. As shown by [32] achieving adequate exposure (in his study it was 10 or more times of hearing the message) is necessary to improve FAS prevention knowledge among the target group. With lower exposure, the massage might be misinterpreted and counterproductive.

All Polish campaigns have been addressed to the general audience what actually means–to nobody. The justification for the need to precisely define the target group and adapt the content and forms of the message to its specific needs can be found on the basis of many, already classical theories from Lasswell model [33] to Prochaska et al. [34]. Based on the qualitative data presented above, it may be proposed to differentiate the FASD prevention messages addressing three groups of women:The key message to women who do not accept even moderate alcohol use during pregnancy should strengthen their attitude (saying: Yes, you are right, keep doing) and, eventually promoting abstinence (or moderate drinking) when planning the pregnancy and/or encouraging use of effective contraceptive measures to avoid unplanned pregnancy.In case of women accepting moderate alcohol use during pregnancy, probably effective would be warning about the health hazard associated with exposing the fetus to moderate doses of alcohol (Like: Is the momentary pleasure of drinking a glass of wine worth worrying about your child’s health for the rest of your life?).Women with alcohol related problems may probably benefit from a campaign based on the CHOICES model [35,36,37] underlying that it’s never too late to stop/moderate your drinking (Although sooner is better) or to prevent getting pregnant. (Just because you have an alcohol problem doesn’t mean you can’t be a good mother. Take the first step and seek support).

The Polish women’s expectations (presented above) suggest at least two theoretical frameworks which might be useful in campaigns planning. First is the exemplification theory [38,39], stating that messages that use concrete, iconic and emotionally arousing exemplars are easily accessed and therefore people tend to rely upon them when making behavioral decisions. And women are especially vulnerable to this kind of messages [40]. Probably the strongest preventive effect would have “testimonials” of biological mothers of children with FASD like ones presented by women from the NOFAS Circle of Hope, who share their stories on the YouTube (https://fasdunited.org/stamp-out-stigma/, accessed on 16 February 2022). Unfortunately, there is no organization in Poland that brings together biological mothers of children with FASD. The fear of stigmatization and social rejection for drinking alcohol during pregnancy is still too strong in our country. A leader as brave, strong and determined to help women who need support to cope with the guilt and grief of drinking alcohol during pregnancy as Kattie Mitchel, the Vice President of the National Organization on Fetal Alcohol Syndrome (NOFAS), has not yet emerged among Polish mothers.

Another theory to guide public health campaign might be the prospect/message framing theory [41,42,43], turning our attention to the fact that every information about a health behavior can emphasize the benefits of taking action (e.g., giving up alcohol during pregnancy increases the chances of your baby’s good health) or the costs of failing to take action (e.g., by drinking alcohol while pregnant you run the risk of having your baby develop FASD). According to this theory the effectiveness of the gain-or loss-framed message is determined by the perceived riskiness of the given behavior–here: not drinking alcohol during pregnancy (as this is the key health behavior concerned in FASD prevention). When discussing prevention of a different health problem, e.g., breast cancer, it is clear that self-examination might be perceived as a risky behavior due to the possibility of detecting a tumor. Therefore, loss-framed appeals should be more persuasive. But in case of other health related behaviors, recommended preventive activity is not associated with any risk, like for example use of sunscreens (it is not unpleasant, not leading to the detection of health problem). And in cases like this, gain-framed appeals should be more persuasive. These practical guidelines from the message framing theory lead to the question how risky is stopping drinking alcohol for women who get pregnant. The answer would be different for women addicted to alcohol and those who drink occasionally. It will be also determined by other social (e.g., alcohol use by a partner, socio-economic situation) or individual (e.g., mental health) factors, what strengthens the need of clearly defined target group.

In the area of FASD prevention, exemplification and message framing theories were tested only by Yu et al. [44]. Their results indicate that exemplar appeals could increase the prevention intention, perceived severity, and perceived fear toward FASD, but only in combination with the loss-framed message. To increase people’s confidence in preventing FASD, messages that contain statistics (not exemplars) and gain-framed appeals, are more effective. The problem is that in this study participated only college students, in great majority not planning pregnancy in the near future and not perceiving FASD as something that could happen to them. Moreover, the history of their alcohol use was not controlled. So the Authors conclude with the same question as formulated above: “Does the fact that avoiding FASD requires the cessation of a (presumably) enjoyable behavior amount to a perceived risk? If so, prospect theory would predict that loss-framed messages would be more effective, as suggested by the results of this study.” (p. 699).

Further support for the loss-framed approach, at least when the target population are educated, middle socioeconomic-class women, who drink alcohol moderately, comes from France et al. [45]. The Authors tested the effectiveness in increasing women’s motivation to abstain from alcohol during pregnancy of four types of messages: (1.) Self-efficacy only—focused on modelling positive behavior and social belonging and acceptance, promoting a sense of self-efficacy in supporting woman’s abstinence from alcohol during pregnancy; (2.) Threat only—focused on a FAS/FASD risk associated with PAE; (3.) Threat and self-efficacy—combining the threat and self-efficacy messages; (4.) Control—a “drink less” advertising without any reference to the pregnancy. The results indicated that advertisements with a threat appeal were more effective at increasing women’s intentions to abstain from alcohol during pregnancy than the self-efficacy message and the control. For future campaigns the Authors recommend the combination of the threat and self-efficacy appeals as less likely to arouse negative emotions and defensive reactions i.e., rejection of the message. The conclusions of the Golińska [29] study of Polish women reasoning about alcohol, pregnancy and needed public health reactions is fully in line with this recommendation.

## 7. Public Health Response to the Belief That There Is no Evidence That Moderate Drinking Is Harmful during Pregnancy

As health professionals knowledge about PAE and FASD is insufficient (e.g., [46,47,48]) it is not surprising that Polish women are confused about the harms to the fetus associated with moderate alcohol use. However, in spite of the differences in the interpretation in terms of appropriate reaction (to drink or not to drink small amounts of alcohol during pregnancy) there is a general consensus in Polish population that there is no evidence of harmful effects of moderate drinking. This claim is challenged by the results of the research published in the last decade.

The results of Lewis et al. [49] and Murray et al. [50] studies may suggest that moderate alcohol consumption in pregnancy is harmful only for the subpopulation of people with very unique variants in genes. PAE is only a mediating factor, which presence is necessary to produce negative outcomes in a child resulting either in the cognitive (IQ scores, analyzed in the first study) or early-onset-persistent conduct (second study) problems. But on the other hand, the same studies may lead to the conclusion that genes associated with developmental disorders in a child are a risk factor only in case of mothers who drank moderately in pregnancy. Among children of nondrinking mothers, the same genotypes are harmless.

Further studies indicated that maternal alcohol use during pregnancy, even at low to moderate levels, is associated with neurological diseases that involve immune-neuroimmune interactions [51]. Low-to-moderate PAE continued across gestation is associated with most negative infant outcomes than moderate-to-high PAE with early reduction [52].

To be honest, the fact that there are also research showing that low/moderate PAE (precisely: up to 1–2 drinks per week or per occasion) might be even good for children’s health [53] has to be noted, too. In this study, boys and girls born to light drinkers were less likely, at age 5, to have high total difficulties scores, hyperactivity and low cognitive test scores compared with those born to mothers who did not drank or drank excessively (what suggests the U-shaped relationship).

But as shown in the systematic review [54] studies indicating that increased alcohol exposure during pregnancy is associated with increased positive mental health in offspring are in the minority. In general, research evidence prove that alcohol use during pregnancy, even at low to moderate levels, is associated with increased risk of mental health problems in the offspring, specifically anxiety/depression, total problems and conduct disorder.

So why it is not known to the general population?

Maybe, because these knowledge is too “fresh”. It is estimated that it takes an average of 17 years for research evidence to reach clinical practice [55]. Probably even more time is needed to reach public awareness. The problem may also be the overly complicated presentation of research results and conclusions, which is not easy to translate into practical tips. For example, it is only in a study showing the positive effects of small amounts of alcohol during pregnancy on a child’s development that the definition of small amounts is formulated in terms of drinks consumed at a specific time [53]. In other studies, small and moderate amounts are defined in statistical analysis, which cannot be easily translated into a language that is understandable to non-specialists [49,50,51,52].

## 8. Conclusions

The purpose of this text was to gather information on how women think about drinking alcohol during pregnancy and to see if public awareness campaigns to reduce the risk of FASD answer their questions and concerns. A systematic review of global research on factors associated with alcohol use, reduction, and abstinence in pregnancy [21] shows that there are relatively few in-depth analyzes explaining why women, not having alcohol problems, drink alcohol during pregnancy. Even fewer such studies have been carried out in Europe, and in Poland, apart from one unpublished study [29], they actually do not exist.

Because without a good understanding of the attitudes, reasoning, fears and beliefs of the target group, there is no question of effective social policy, it is obvious that as much research as possible is needed. However, even so, they are only an introduction to planning preventive and intervention measures. Another important element of the process should be testing the proposed methods of communication with recipients. Examples of such research in the area of FASD can be found in the USA [44] or Australia [45], but not in Poland. Undoubtedly, the public sector can learn a lot from the marketing effects of commercial companies.

Among International Standards on Drug Use Prevention [31] there is also one recommendation not discussed above that media campaigns should be “strongly connect with other existing drug prevention programmes in the home, school and community”. This is undeniably true, but the analysis of what services are already available in Poland and which are still missing, goes beyond the scope of this study. Some information on the available offer for pregnant women with alcohol problems and people with FASD has been presented elsewhere [30]. And they confirm how much remains to be done to effectively reduce the risk of drinking alcohol during pregnancy.

In accordance with the recommendations of the World Health Organization policy measures diminishing economic and physical availability of alcohol could be helpful to reduce alcohol consumption and FASD risk in Poland [56,57]. However, taking in to account Poles’ reluctance to restrictive solutions [14] introduction of strong alcohol-control policies might be difficult. Probably, a good strategy may be, as suggested by Lesch & McCambridge [58] not attempting to persuade decision-makers and/or opponents with evidence and argumentation but building the strong and broad coalition appealing for the re-framing of policies e.g., by introducing mandatory warning labels on alcoholic products or banning alcohol advertising. The first policy seems to be highly acceptable in Polish society (87% of positive opinions) while the second, although probably even more effective, is supported by 55% of Poles, only [59]. However, there is no doubt that all activities in the field of social communication, prevention and support for women at high risk of PAE should be strengthened by consistently introduced political solutions.

## Figures and Tables

**Table 1 ijerph-19-04552-t001:** Inclusion criteria for each focus group.

Research Design	Reject Alcohol Use during Pregnancy	Think That (Total) Abstinence from Alcohol during Pregnancy is not Necessary
Pregnant women	FGI 1.	FGI 3.
Not pregnant women	FGI 2.	FGI 4.

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
