# Peer review of "Is Public Health Response to the Phenomenon of Alcohol Use during Pregnancy Adequate to the Polish Women’s Needs?"

_ijerph, 2022, doi:10.3390/ijerph19084552_

Round 1

Reviewer 1 Report

Please modify the conclusion in the end of the article and put forward the author's opinions to provide scientific control policy of alchol.

Author Response

Thank you for the review. The paragraph concerning the possibility of introducing changes to the alcohol policy in Poland is added to the conclusions.

Reviewer 2 Report

Comments

This review is focused on gathering information on women’s attitude for drinking alcohol during pregnancy, and trying to find effective methods to activate public awareness campaigns to reduce the risk of FASD by controlling or giving up alcohol uptake during pregnancy. And this paper especially concerned about polish women’s response to drink alcohol, pregnancy and FAS/FASD.

Major comments:

  1. This paper is in general well written. However, I think it needs more data to support their findings;
  2. In the future, if they broad their research to different races, they may find some more valuable results.

Author Response

Thank you for the review. As this paper "especially concerns Polish women's response to  drink alcohol, pregnancy and FAS/FASD", according to my knowledge, adding more data collected in the country of interest (Poland) is not possible.

Broadening this kind of research to different races would certainly be interesting and valuable, but it is beyond the scope of this paper. However, following the recommendations of other reviewers, I emphasized in the title and in the text that it focuses on the situation in Poland.

Reviewer 3 Report

Many thanks for giving me the opportunity to review the manuscript entitled "Is public health response to the phenomenon of alcohol use during pregnancy adequate to the women’s needs?"

The article is a comprehensive review of the phenomenon of alcohol use during pregnancy. It has two goals. The first goal of this article is to analyze the current state of knowledge about the reasons for drinking alcohol, especially moderate drinking, during pregnancy. The considerations are based mainly on the results of foreign and Polish qualitative research. The second goal is to look at the activities in the field of public education carried out in Poland so far, in terms of their adequacy to the needs in the field of prevention of prenatal alcohol exposure and fetal alcohol spectrum disorders. Extra these goals, the author has formulated recommendations for future activities in this area to achieve effective interventions through the media.

I was curious to read this article. It lacks numbers and statistics, and provides deep and sensitive insight into the problem. It can be seen that the author is well aware of the problem at hand. I hope this article will be interesting to other readers as well. And not just for those professionals who work in the field of pregnancy care. The information in this article may be useful to public health professionals developing strategies for intervention in the population. The article can also be used as educational material for students preparing to become public health professionals.

The article is written in a typical format. I did not notice any significant shortcomings with regard to the material presented in the article. Nevertheless, while reading the submitted manuscript, several inaccuracies were noticed, which I recommend to fix:

  1. Introduction, Line 36: correct as follows: "... known as Fetal Alcohol Spectrum Disorder (FASD)."
  2. Introduction, Lines 41-42: correct as follows: "... who, for instance, in US constitute about 5% of the pregnant population [12], ..."

Thank you for considering my opinion. I encourage authors to keep on working in the field.

Author Response

Thank you. All proposed changes have been made.

Reviewer 4 Report

The paper sounds really interesting but needs to improve some aspetcs.

  1. The title do not let understand that major focus of the paper regards Polish women. It is necessary to add this information in the title. On the other hand the manuscript claims to have the character of a review but very poor information is reported about the alcohol use during pregnancyin in "other" countries. In line 47 the authors report the word "foreign" countries added to the word "polish": it lead to think that the manscript regards polish context.
  2. In materials and methods, the Authors have to stress more about why the polish context offer a good environment where the phenomenon of alcohol use during pregnancy could be studied.

Author Response

Thank you for the review. In accordance with your comments, I modified the title and added the explanation why the polish context offers a good environment where the phenomenon of alcohol use during pregnancy could be studied

Round 2

Reviewer 4 Report

Very good work. Thank you.